# Influence of Warm Mix Asphalt Additives on the Physical Characteristics of Crumb Rubber Asphalt Binders

Munder Bilema [1,2,*] , Choon Wah Yuen [1,2] , Mohammad Alharthai [3,4] , Zaid Hazim Al-Saffar [5],
Salam Ridha Oleiwi Aletba [6] and Nur Izzi Md Yusoff [7]

[1] Centre for Transportation Research, Universiti Malaya, Kuala Lumpur 50603, Malaysia; yuencw@um.edu.my
[2] Department of Civil Engineering, Faculty of Engineering, Universiti Malaya, Kuala Lumpur 50603, Malaysia
[3] Department of Civil Engineering, Najran University, Najran 66462, Saudi Arabia; maalharthai@nu.edu.sa
[4] Science and Engineering Research Center, Najran University, Najran 66462, Saudi Arabia
[5] Building and Construction Engineering Department, Engineering Technical College of Mosul,
Northern Technical University, Mosul 41002, Iraq; zaid.alsaffar@ntu.edu.iq
[6] Civil Engineering Department, University of Technology-Iraq, Alsina'a Street, Baghdad 10066, Iraq;
40372@uotechnology.edu.iq
[7] Department of Civil Engineering, Universiti Kebangsaan Malaysia, Bangi 43600, Selangor, Malaysia;
izzi@ukm.edu.my
[*] Correspondence: munderbilema@um.edu.my or munderbilema@gmail.com

**Abstract:** This investigation is centered around the application of warm mix asphalt (WMA) technologies to address workability concerns linked to rubberized asphalt binders. The primary aim of incorporating crumb rubber (CR) and WMA additives is to establish a robust paving method that fosters energy conservation, efficient waste management, noise reduction, and improved overall performance. The current study aims to comprehensively characterize and differentiate the physical attributes of rubberized asphalt binders by employing three distinct WMA additives: Sasobit, Cecabase RT and Rediset WMX. These additives are introduced into eight unique asphalt binders. Laboratory assessments are carried out to evaluate the workability and physical properties of these binders. The evaluation encompasses penetration, softening point, penetration index, penetration viscosity number, storage stability, ductility, viscosity, and stiffness modulus analyses. The findings indicate that the rubberized asphalt binder enhanced with Sasobit demonstrates the highest levels of both hardness and softening point in comparison to asphalt binders supplemented with alternative WMA additives. The evaluation of storage stability underscores the satisfactory stability across all modified asphalt binders. Both the unmodified and modified binders meet the requirements stipulated by the ductility test; the rubberized asphalt binder modified with Rediset falls short. The rubberized asphalt binder improved with Sasobit displays the most notable enhancement in workability. Furthermore, the blend of crumb rubber and Sasobit binder reveals the highest stiffness modulus values under conditions of intermediate and high temperatures with 1.88 and 0.46 MPa, respectively. In summation, the rubberized asphalt binder incorporating crumb rubber with Sasobit showcases superior improvements in both stiffness and workability compared to counterparts modified with Cecabase RT and Rediset WMX.

**Keywords:** warm mix asphalt (WMA); crumb rubber (CR); WMA additives; physical properties; workability; stiffness modulus

## 1. Introduction

Authorities across the globe are dealing with limited budgets for constructing and maintaining infrastructures by implementing cost-saving strategies that utilize recycled materials [1,2]. Pavement researchers are still looking for cost-effective solutions that reduce project expenditures by incorporating waste material into road pavements [3]. Several studies have examined the performances of pavements modified with waste materials,

including recycled asphalt shingles (RAS), reclaimed asphalt pavement (RAP), glass, steel slag, waste frying oil (WFO), waste engine oil (WEO), and crumb rubber (CR) [4–7].

The rapid growth of the automotive sector and the number of vehicles on the road each year has increased the demand for tires. Globally, 1.5 billion tires are produced, and 4.0 billion tires are disposed of annually [8]. The decomposition of discarded tires, sometimes known as black pollution, could be worse than plastic or white pollution. The thermosetting nature of natural and synthetic rubbers, which seldom degrade in typical climate conditions, makes scrap tire disposal challenging. The conventional methods for disposing of waste tires are landfilling and burning. The landfilling of scrap tires poses health risks and can cause unintentional fires, while burning scrap tires pollutes the environment. In short, waste tire dumping causes environmental, economic, health, and social issues [9]. Recycling scrap tires in asphalt pavements is a cost-effective and environmentally responsible approach to disposing of these waste materials [10,11]. The usage of crumb rubber (CR) is advantageous for environmental reasons (i.e., noise isolation). The exposure to emissions in asphalt-rubber production was the same as in traditional asphalt production. Moreover, in comparison to the influence of other factors, the effect of CRM on emissions may be minimal. The dryer's fueling rate, mix temperature, asphalt throughput rate, and binder content are some of those factors [12].

The wet and dry methods exist for integrating crumb rubber into asphalt pavements [13,14]. Wet bitumen modification techniques involve blending the asphalt binder and CR at temperatures between 160 °C and 200 °C, adhering to specific shear rates and durations [15]. At these elevated temperatures, crumb rubber particles rapidly expand due to the absorption of the lighter components of the bitumen during mixing [16]. The incorporation of crumb rubber into asphalt binders significantly enhances the characteristics of the modified binder. Various factors, including raw material properties and interaction conditions, exert an influence on the properties of rubberized binders. Researchers have extensively employed CR to ameliorate the performance of rubberized asphalt pavements while upholding environmental considerations [17]. CR-modified asphalt binders exhibit augmented viscosity, heightened resistance to rutting, cracking, and moisture susceptibility, as well as elevated manufacturing and laying temperatures (approximately 10 °C) in comparison to conventional blends [18]. However, challenges associated with CR-modified asphalt pavements encompass reduced workability and heightened temperature prerequisites at asphalt manufacturing facilities [19]. Recent studies have concentrated on mitigating production and laying temperatures and enhancing workability. On the other hand, the absence of a universally standardized warm mix asphalt (WMA) mix design underscores the necessity to comprehend the fundamental mechanisms through which additives operate within the mixture. The insufficiency of information concerning the capabilities and dynamics of WMA, prevalent in various global regions, is attributed to factors such as the variation in WMA additives, suggested quantities, and technological approaches. This knowledge gap can be attributed in part to the limited understanding of WMA additives, which impedes the cultivation of expertise in WMA practices among industry professionals [20].

In recent years, warm mix asphalt has been gaining popularity due to its benefits, including reduced fossil fuel consumption, reduced greenhouse gas emissions, lower manufacturing, and laying temperatures [20]. Many studies and field cases have demonstrated the eco-friendliness, superior performance, and economic benefits of WMA technologies, such as improved compaction and mixing temperatures, longer hauling distances, and paving workability [18,21], which contributed to the rapid increase of its market share. In the United States, WMA asphalt made up about 38.9% of the total paving, the equivalent of 147.4 million tons in 2017 [22]. There was a 26% increase in WMA output between 2016 and 2017. The production in Europe, Japan, Canada, and South Africa is still insufficient. The data from the European Asphalt Pavement Association showed that, between 2013 and 2017, several European countries allowed the use of WMA, which leads to increasing use of WMA technology year after year [18].

In recent studies, three distinct WMA techniques have been devised: (a) utilization of foaming additives, (b) incorporation of chemical additives (surfactants), and (c) inclusion of non-foaming additives. Recent examinations conducted in both laboratory settings and in the field have demonstrated that pavements constructed using WMA exhibit equivalent or improved performance compared to those constructed using hot mix asphalt (HMA) [23]. Researchers have observed that the introduction of additives for WMA into asphalt blends leads to a notable reduction in the temperatures required for mixing and compaction, with temperature decreases ranging between 20 °C and 40 °C [19,24]. WMA technologies that employ fluidifying additives to lower the viscosity of bitumen during mixture production temperatures, while upholding bitumen functionality, offer a potential resolution to challenges linked with crumb rubber-modified (CRM) binders. Incorporating WMA additives into CR asphalt binders has shown the potential to decrease mixing and compaction temperatures by 15 to 30 °C [25,26]. Numerous WMA additives have demonstrated the capacity to lower the production temperature of rubberized binder mixtures by reducing viscosity and enhancing the workability of the rubberized binder [27,28]. A sustainable paving approach has been developed by researchers, involving the use of WMA asphalt to modify rubberized asphalt binders, thereby contributing to environmental preservation and performance enhancement [29]. The integration of Sasobit into the rubberized binder leads to a decrease in the high-temperature viscosity and low-temperature ductility of rubber asphalt, while also resulting in an increase in the softening point. This combination of effects is beneficial for enhancing workability and promoting high-temperature stability [30]. In a prior investigation conducted by Sesay et al. [31], a comparative analysis was carried out involving three distinct WMA additives: Sasobit, Rediset, and Alube. These additives were introduced into the rubberized asphalt binder to assess their influence on high and intermediate temperature performance. The study revealed that the rubberized asphalt binder incorporating Sasobit exhibited superior high-temperature performance in comparison to the rubberized asphalt binder with Alube. Conversely, concerning intermediate-temperature performance, the rubberized asphalt binder with Rediset displayed the most favorable outcomes.

The available literature reveals a dearth of investigations concerning the collective impact of various CR asphalt binders modified with optimal quantities of WMA additives. Addressing this gap, the present study aims to investigate the physical attributes of diverse CR binders, each modified with Sasobit, Rediset, and Cecabase. The aim of this investigation is to identify a suitable WMA additive that enhances the workability of the rubberized binder while maintaining a stiffness value higher than that of the original asphalt binder.

## 2. Material and Methods

### 2.1. Asphalt Binder

The bitumen binder employed in this research possesses a 60/70 penetration grade and is supplied by the Kemaman Bitumen Company Sdn. Bhd, Selangor, Malaysia. This binder boasts a flash point of 240 °C and a density of 1.02 gm/cm$^3$.

### 2.2. Crumb Rubber

The Malaysian supplier, Miroad Rubber Industries Sdn Bhd, furnishes crumb rubber powder with a mesh size of 20 (passing 0.15 mm). The crumb rubber modifier (CRM) is manufactured through a process of mechanical shredding followed by grinding at ambient temperature.

### 2.3. WMA Additives

This research used three warm mix additives, an organic additive (Sasobit), and two chemical additives (Rediset WMX and Cacebase). The dosage of the WMA additives followed the recommendations of previous studies [32–34]. Figure 1 shows all warm mix asphalt additives and crumb rubber used in this study. Table 1 shows the physical characteristics of the warm mix asphalt additives used in this research.

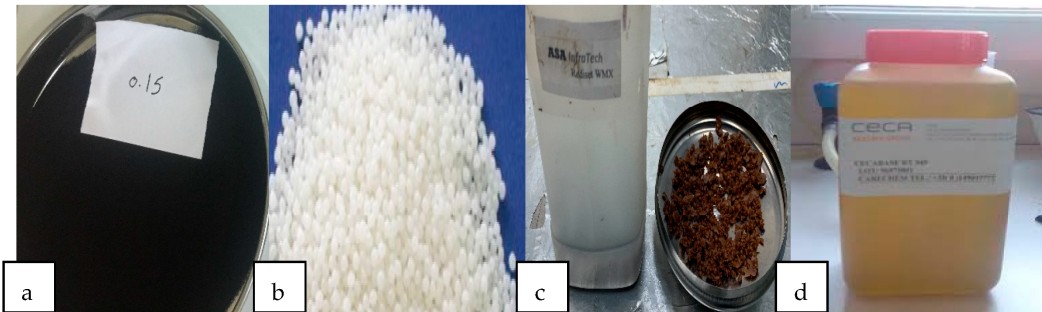

**Figure 1.** The additives used in this research. (**a**) Crumb rubber. (**b**) Sasobit. (**c**) Rediset WMX. (**d**) Cecabase.

**Table 1.** Properties of the WMA additives.

| Properties | Sasobit | Cecabase RT 975 | Rediset WMX |
|---|---|---|---|
| Appearance at 25 °C | Pastille flakes | Liquid | Pastilles |
| Density at 25 °C (g/cm$^3$) | 0.9 | 0.997 | 0.55 |
| Color | Off-white | Dark amber liquid | Light brown |
| Melting point (°C) | 100–120 | - | 80–90 |
| Flash point (°C) | 285 | >200 | 253 |

### 2.3.1. Sasobit

Sasobit, among the pioneering WMA additives, was introduced for pavement production. Its inaugural field trial occurred in Hamburg, Germany in 1997. Sasobit, a synthetic paraffin wax, is synthesized using the Fischer–Tropsch technique, involving the combination of heated coal and natural gas with steam and a catalyst [18]. Its viscosity surpasses that of the binder below its melting point. The manufacturer recommends a dosage range of 0.8% to 3% by weight of the binder [35].

### 2.3.2. Cecabase RT 975

Cecabase RT 975 is a chemical additive produced by the Arkema Group in France. It is liquid at 25 °C and can reduce the asphalt mixing and laydown temperatures by 20–40 °C. Cecabase RT 975 is one of the most popular liquid WMA additives with a recommended dose of 0.2–0.5% and does not require curing in the mixing process. It is pumped directly into the asphalt mixture or mixed with the asphalt binder [36].

### 2.3.3. Rediset WMX

Rediset WMX is a blend of organic additives and surface-active chemicals produced by a Dutch firm, Akzo Nobel. Rediset WMX was first proposed in 2007 to resolve the reported flaws in the warm mix asphalts, including the effects of water on warm asphalt mixes and their lower hardness compared to hot mixes, and the unpredictability of their low-temperature characteristics. Rediset WMX is mixed with the asphalt binder or mixture at a dosage of 0.5 to 2.5% by weight of the asphalt binder [34].

### 2.4. Preparation of the CR Binders Containing the WMA Additives

The virgin asphalt binders were placed in a container and heated in a 110 °C oven for 60 min. The asphalt binder was added with 5% CR for 30 min at 700 rpm and 177 °C. Then, the hot plate was left for five minutes to reach the desired temperature before the next addition, which adjusted the temperature to 120 °C [32]. The required amounts of the WMA additives were added at 1000 rpm and 120 °C for 10 min to ensure that the asphalt binders were sufficiently fluid, uniform, and consistent [37]. The modified binders were placed in the containers for investigation. Table 2 presents the composition of all asphalt binders, and Figure 2 shows the high-shear mixer used to mix the modified asphalt binders.

**Table 2.** Composition of the modified and virgin asphalt binders.

| Binder | ID | Asphalt Binder (%) | Crumb (%) | Warm Mix Additive (%) |
|---|---|---|---|---|
| Virgin 60/70 | V | 100 | - | - |
| Crumb rubber | CR | 100 | 5 | - |
| Sasobit | SA | 100 | - | 1.5 |
| Cecabase | CE | 100 | - | 0.44 |
| Rediset | RE | 100 | - | 1.5 |
| Sasobit + crumb rubber | SECR | 100 | 5 | 1.5 |
| Cecabase + crumb rubber | CECR | 100 | 5 | 0.44 |
| Rediset + crumb rubber | RECR | 100 | 5 | 1.5 |

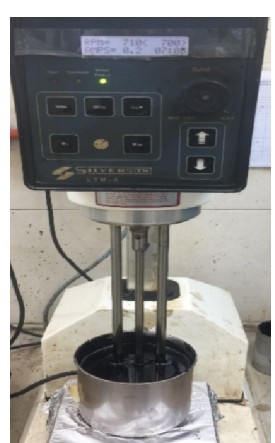

**Figure 2.** The high-shear mixer for blending the modified asphalt binders.

*2.5. Testing Program*

Table 3 summarizes the penetration, softening point, penetration index, storage stability, ductility, loss on heating, and viscosity tests performed in this research to determine the properties of the conventional asphalt binder. The softening point and penetration values were used to compute the penetration index (*PI*) using Formula (1).

$$PI = \frac{1952 - 500 loglog\ PEN - 20SP}{50 loglog\ PEN - SP - 120}, \tag{1}$$

where *PI* is the penetration index, *PEN* is the penetration value (0.1 mm), and *SP* is the softening point value (°C). The viscosity and penetration values were used to compute the penetration viscosity number (*PVN*) using Formula (2).

$$PVN = \frac{4.258 - 0.7967\ log\ P - loglog\ V}{(0.795 - 0.1858\ log\ P)} \tag{2}$$

where *PVN* is the penetration viscosity number, *V* is the viscosity in centistokes measured at 135 °C (cp), and *P* is the penetration value at 25 °C (0.1 mm).

**Table 3.** The physical properties tests conducted in this research.

| Asphalt Binder Test (Unit) | Test Standard | Specification |
|---|---|---|
| Penetration at 25 °C (0.1 mm) | ASTM D5 | 60–70 |
| Softening point (°C) | ASTM D36 | 48–52 |
| Penetration index | - | −2 to 2 |
| Penetration viscosity number | - | - |
| Storage stability (°C) | ASTM D5892 | <2.2 |
| Ductility at 25 °C (cm) | ASTM D113 | Min. 100 |
| Loss on heating (%) | AASHTO T240 | Max. 1.00 |
| Viscosity at 120 and 135 °C (cp) | ASTM D4402 | - |
| Stiffness modulus (MPa) | - | - |

Van der Poel [38] introduced the concept of stiffness modulus, which represents the ratio between a consistent uniaxial stress and the resultant uniaxial strain at a specified time. By examining the outcomes of tests conducted on 47 bitumens, Van der Poel constructed a graphical tool, known as a nomograph, designed to ascertain the stiffness of asphalt binder, accounting for temperature and loading time (or frequency), given knowledge of the softening temperature and penetration. For the practical determination of binder stiffness, the BitProps program is a useful tool. This program utilizes a digitized version of Van der Poel's nomograph, developed by G. Rowe and M. Sharrock [39], facilitating the ease of calculation. The study evaluated the stiffness characteristics of eight distinct asphalt binders, both unmodified and modified, across varying temperature conditions: low, intermediate, and high. This assessment employed five distinct loading scenarios, considering outcomes from penetration and softening point tests. The selected temperatures for this investigation corresponded to the low, intermediate, and high temperature ranges observed in Kuala Lumpur, Malaysia. Figure 3 shows the flowchart of the research.

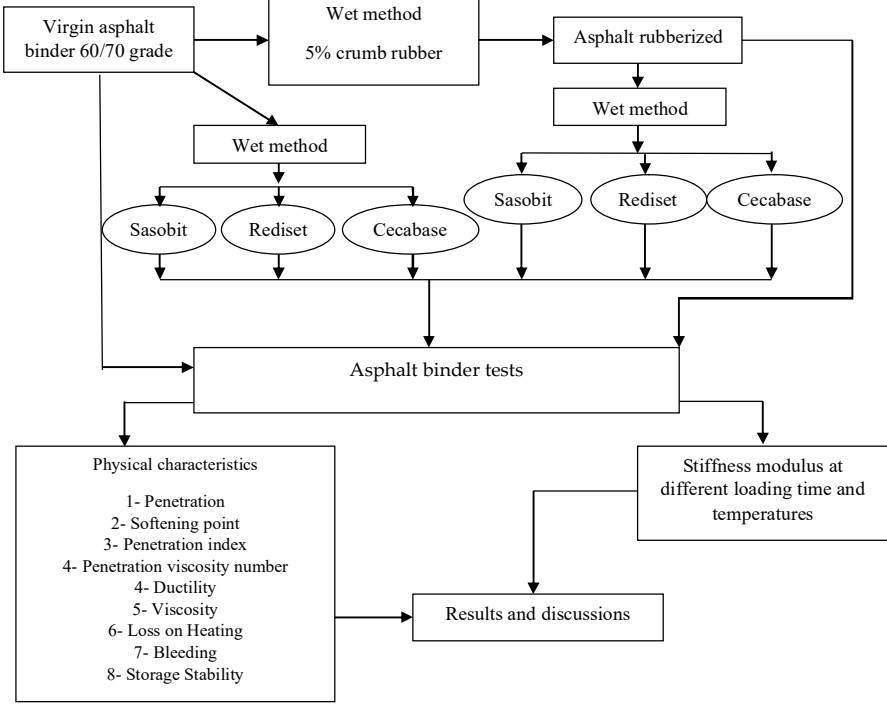

**Figure 3.** The flowchart of the study.

## 3. Results and Discussion

### 3.1. Penetration

The assessment of asphalt binder characteristics involves multiple tests to determine their properties. Figure 4 illustrates the penetration values of various asphalt binders at 25 °C, which serve as an indicator of their hardness or softness. A lower penetration value in CR asphalt binders corresponds to increased hardness. Previous research [40] highlighted that introducing crumb rubber into asphalt binders led to a significant reduction in penetration values, contingent on CR content. Among WMA additives, Cecabase exhibited minimal impact on asphalt binder hardness, with the modified binder's penetration value aligning closely with that of the virgin binder. This was confirmed by Awazhar et al. [41], who noted Cecabase's limited influence on penetration values. In contrast, Sasobit and Rediset exerted more substantial effects on asphalt binder hardness than Cecabase. The liquid form of Cecabase had minimal impact on the hardness of the CECR asphalt binder. The SECR asphalt binder demonstrated the lowest penetration value due to the combined influence of Sasobit and crumb rubber, which augmented the binder's hardness. Bilema et al. [37] reported comparable findings regarding the combined effect of crumb rubber and Sasobit on penetration values, consistent with our current study.

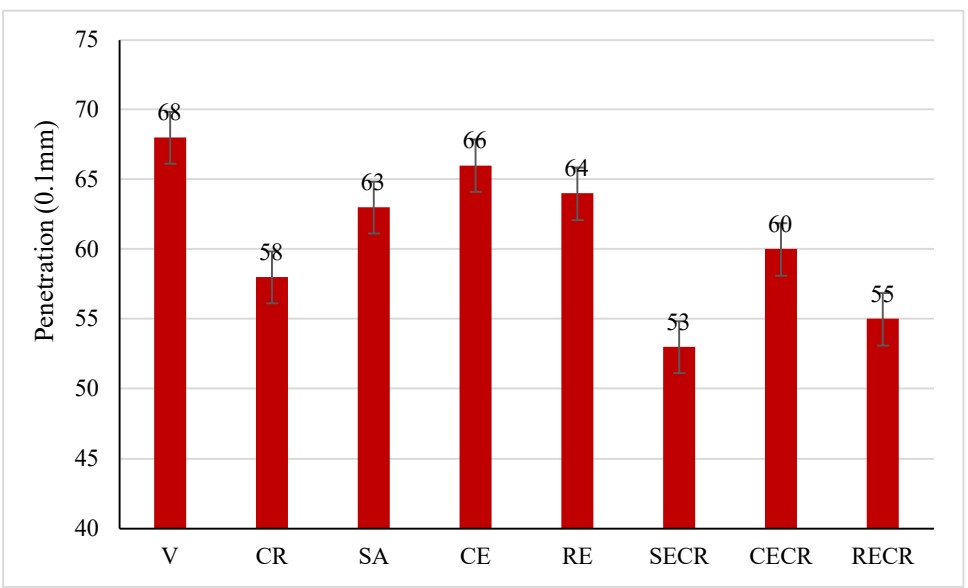

**Figure 4.** The penetration of the asphalt binders.

### 3.2. Softening Point

The outcome of the softening point test, presented in Figure 5, reveals the temperature at which asphalt binders transition from semi-solid to soft states. Higher softening points are preferable for greater resilience in warm conditions. CR-modified asphalt binders exhibited higher penetration values, aligning with previous research [42]. Among WMA additives, Cecabase demonstrated the lowest softening point (47.5 °C), while Sasobit exhibited the highest (50.5 °C). The CECR asphalt binder displayed a relatively low softening point, whereas the SECR binder exhibited the highest value (54 °C), attributed to the combined influence of crumb rubber and Sasobit, resulting in higher melting temperatures. Kök et al. [43] obtained similar results, concluding that Sasobit enhanced softening points. In a preceding study undertaken by Akpolat [44], it was determined that the incorporation of Sasobit additive in combination with CR leads to an elevation in the value of softening points. This concurs with the findings observed in the current research.

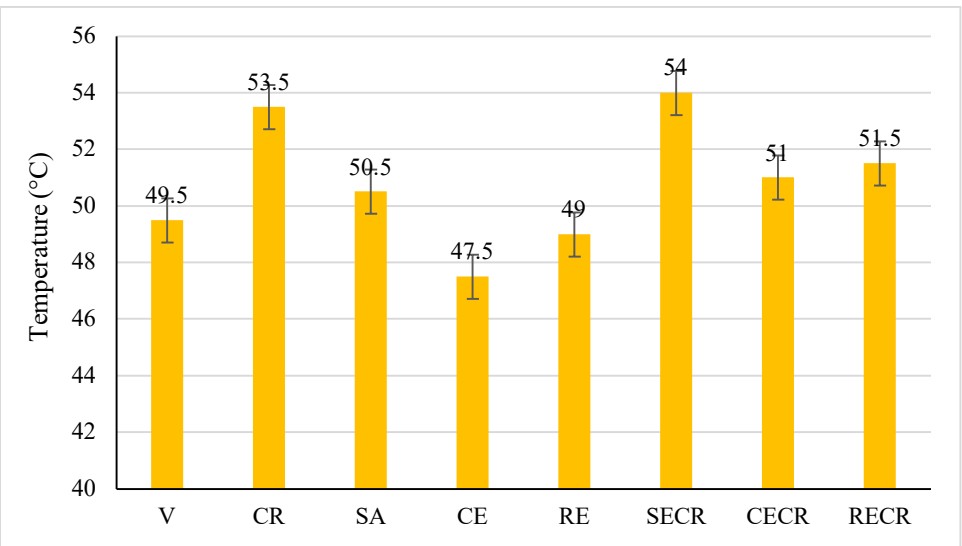

**Figure 5.** The softening point result of the asphalt binders.

### 3.3. Viscosity

The Brookfield rotational viscosity test gauges asphalt binder workability at temperatures of 120 and 135 °C, determining their capacity to withstand shear forces. Figure 6 displays viscosities for virgin and modified asphalt binders. At the higher test temperature of 135 °C, asphalt binders demonstrated reduced viscosity values. CR asphalt binders exhibited the highest viscosity values at both temperatures, attributed to crumb rubber particle behavior. Elevated viscosity of aged binders can decline asphalt mixture workability, causing reduced adhesion between aggregates and binders [45]. Higher viscosity demands elevated laying, compaction, and mixing temperatures, consuming more energy, as noted by Sengoz and Isikyakar [46]. WMA additives lowered viscosities across asphalt binders. Sasobit had the most significant influence on binder workability, whereas Rediset had a limited effect. Viscosity in all asphalt binders, with crumb rubber and WMA additives, was akin to virgin asphalt binders. SECR asphalt binder exhibited the lowest viscosity values at both temperatures due to Sasobit's substantial workability enhancement, consistent with Veeraiah and Nagabhushanarao's findings [47].

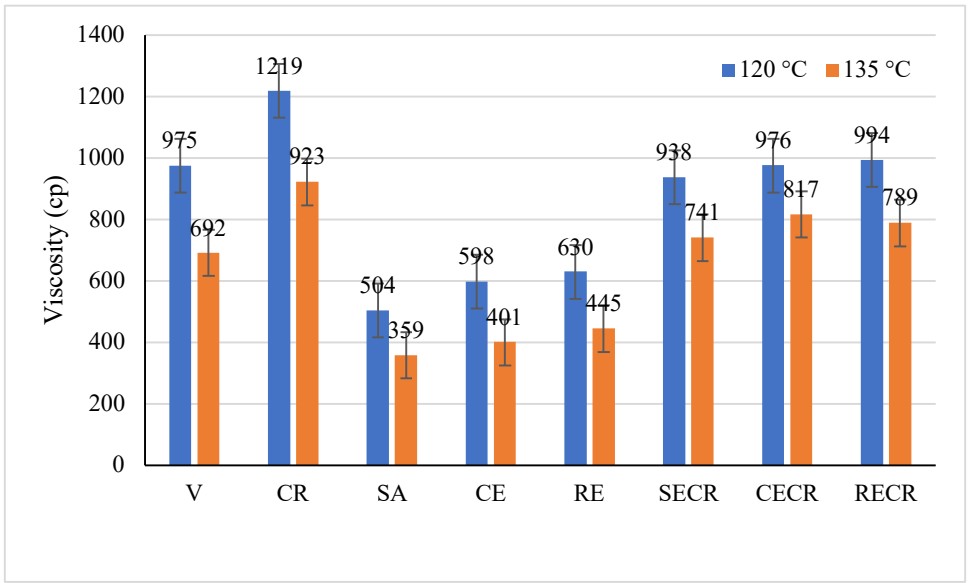

**Figure 6.** The viscosity of the asphalt binders at 120 and 135 °C.

### 3.4. Temperature Susceptibility

Temperature susceptibility of modified asphalt binders was assessed using the penetration index (*PI*) and penetration viscosity number (*PVN*) based on penetration and softening point at 25 °C and viscosity at 135 °C. Higher *PI* and *PVN* values indicated reduced temperature susceptibility. *PI* classified asphalt binder type and suitability for highway construction, while *PVN* gauged temperature susceptibility. The findings in Table 4 indicate that both *PI* and *PVN* values increased with crumb rubber addition, signifying lower temperature susceptibility in CR asphalt binders. Except for Sasobit, WMA additives displayed similar behavior in raising temperature susceptibility values. Asphalt binders modified with WMA additives and crumb rubber exhibited temperature susceptibility akin to virgin asphalt binders. The utilization of Sasobit in the rubberized asphalt binder leads to an elevation in the PI value, accompanied by a reduction in the *PVN* value. This aligns with the outcomes observed in the present study [48]. Table 4 shows the outcome of the *PVN* and *PI* of all asphalt binders in the study.

**Table 4.** The PI and PVN for all asphalt binders.

| ID | PI | PVN |
|----|-----|------|
| V | −0.54 | −0.06 |
| CR | 0.04 | 0.39 |
| SA | −0.49 | −0.92 |
| CE | −1.17 | −0.69 |
| RE | −0.83 | −0.59 |
| SECR | −0.02 | 0.004 |
| CECR | −0.48 | 0.23 |
| RECR | −0.62 | 0.07 |

### 3.5. Storage Stability

The storage stability test, depicted in Figure 7, was conducted to assess the success of binder modification and its stability during storage. Acceptable storage stability demands a minimal temperature difference between sample top and bottom. All asphalt binders showed adequate stability without significant separation, owing to slight variations in softening points within 2.2 °C. Virgin and WMA-modified asphalt binders exhibited lower storage stability, with a 0.5 °C difference. RECR asphalt binder exhibited the highest storage asphalt due to the crumb rubber's contribution. Elevated crumb rubber content decreased modified binder stability, aligning with prior studies [21].

### 3.6. Ductility

The outcome of the ductility test, depicted in Figure 8, illustrates the evaluation of asphalt binder flexibility and deformation. The figure indicates that the inclusion of CR resulted in a decrease in asphalt binder ductility. This reduction in ductility was attributed to the absorption of the asphalt binder's oily components by the rubber powder, leading to an increase in the mass of rubber particles. Consequently, the CR asphalt binder exhibited a greater thickness compared to unmodified asphalt binder samples, aligning with the observations made by Mashaan et al. [49]. Across all asphalt binders, the introduction of WMA additives brought about a reduction in ductility. Sasobit had a minimal effect on asphalt binder ductility, while Rediset produced the highest difference in the ductility values. Sedaghat et al. [50] reported that Sasobit reduced the ductility value, which is consistent with the result of this research. The 109 and 101 cm ductility values of the SECR and CECR asphalt binder fulfilled the ductility test requirement. However, the RECR asphalt binder did not meet the ductility test requirement of 98 cm.

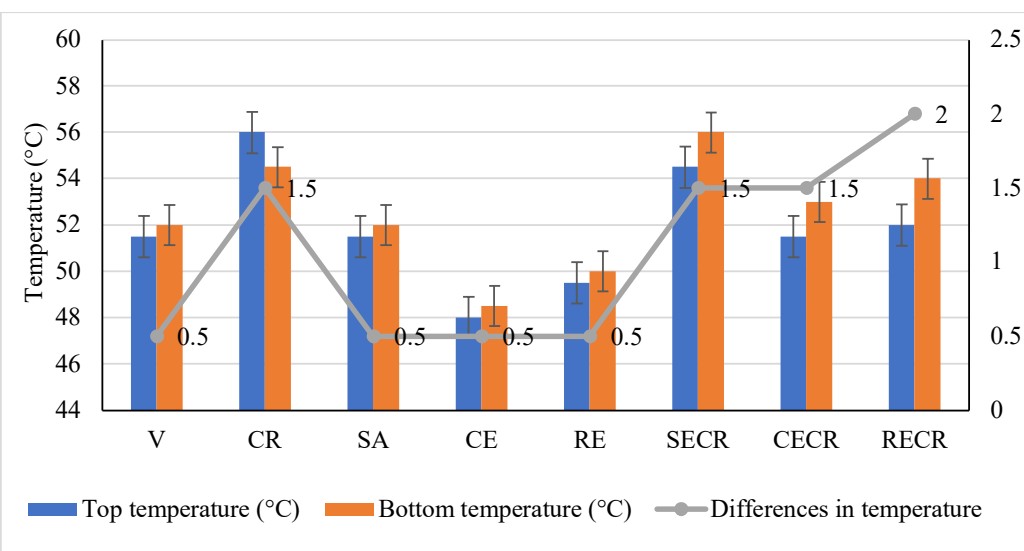

**Figure 7.** The storage stability of the asphalt binders.

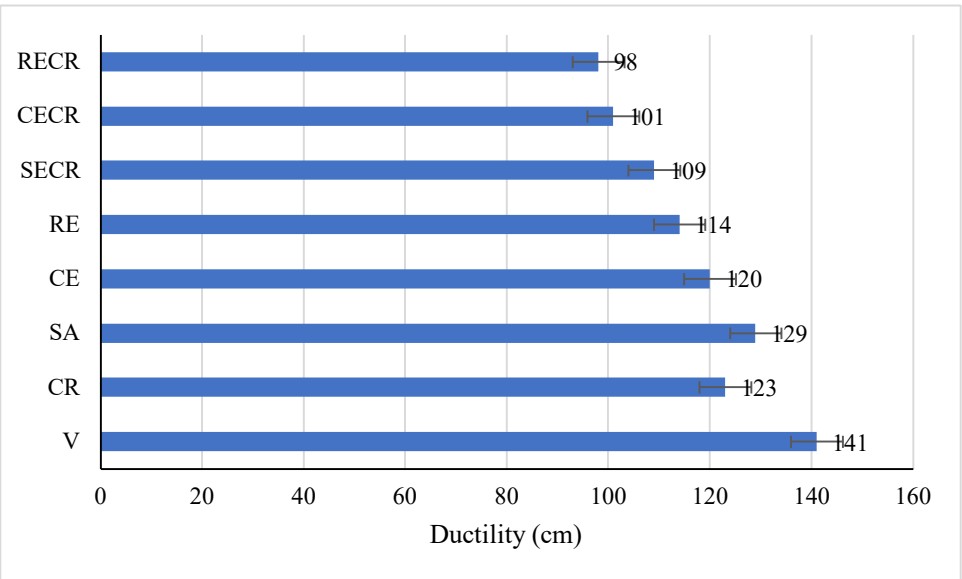

**Figure 8.** The ductility of the asphalt binders.

### *3.7. Loss on Heating*

This research conducted the Rolling Thin Film Oven (RTFO) test to determine the mass loss on heating when the samples were exposed to high temperatures and pressures. The impurities and particles in the asphalt binders became volatile and escaped into the air at high temperatures, causing the asphalt binder to lose some mass. Loss on heating is a critical factor since a considerable loss on heating can cause problems with the asphalt binder's workability. The loss on heating typically ranges from 0.05 to 0.5%, and the maximum loss on heating is 1.0%. Figure 9 presents the loss on heating for all asphalt binders. All asphalt binders in this study fulfilled the 0.05 to 0.5% requirement for the loss on heating test and showed a lower loss on heating than the virgin asphalt binder since the blending process exposed them to a high temperature for a particular period. In a prior investigation conducted by Li et al. [48], it was determined that various dosages of rubberized asphalt binder and Sasobit yielded comparable outcomes to the original asphalt binder, as the percentages fell within the designated test range. These findings correspond harmoniously with the results observed in the current study.

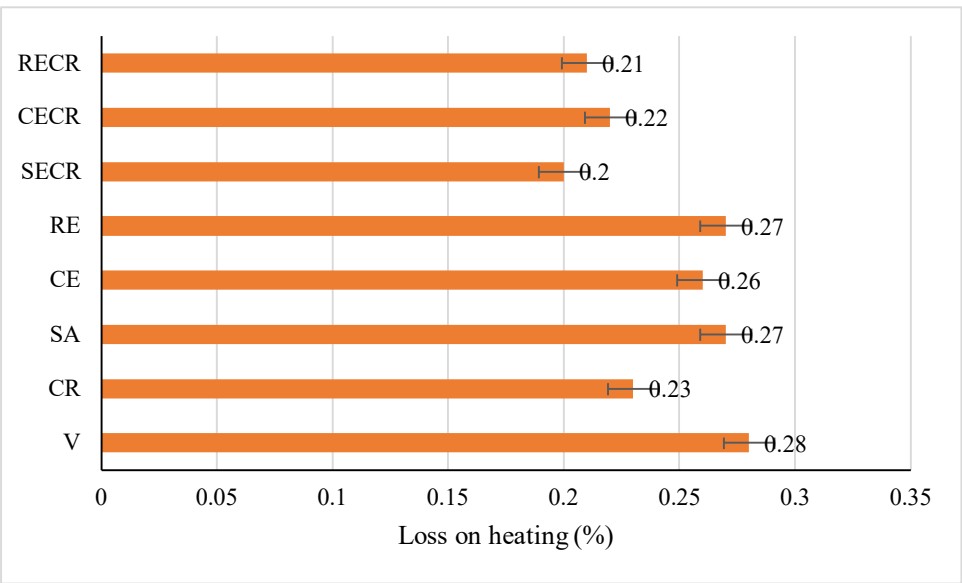

**Figure 9.** The loss on heating for the asphalt binders.

*3.8. Stiffness Modulus*

Table 5 presents two predominant patterns: as test temperatures rise, stiffness values decline, whereas stiffness values increase as loading time decreases. Notably, at a loading time of 10 s, the SECR exhibits the most elevated stiffness modulus figures under intermediate and low temperatures, measuring 0.07 and 0.49 MPa, respectively. Conversely, the CE displays the least stiffness among all unmodified and modified asphalt binders at intermediate temperatures, registering 0.03 MPa. Similar trends in stiffness modulus emerge at loading times of 5, 1, and 0.5 s, mirroring the observations at a 10 s loading time. At the lowest loading times and temperatures explored in this study, modified asphalt binders manifest distinct behavior. Specifically, the RECR demonstrates the highest stiffness values at low temperatures, reaching 11.31 MPa. Among the warm mix asphalt additives without crumb rubber, Sasobit records the highest stiffness values across all temperatures and loading times, except at low temperatures and loading times of 0.5 and 0.1 s, where stiffness values for CE measure 3.23 and 9.93 MPa, respectively. Comparable stiffness values are observed between RECR and SECR. The influence of loading time on stiffness values is evident from the data in Table 5, revealing a notable impact. Reducing the loading time enables a deeper exploration of asphalt binder stiffness behavior. The presence of crumb rubber yields higher stiffness values compared to warm mix asphalt additives, attributed to the combination of low penetration and elevated softening point values. Table 5 shows stiffness modulus values for virgin and modified asphalt binders at different loading times and temperatures.

**Table 5.** The stiffness modulus for unmodified and modified asphalt binders.

| Asphalt ID | Stiffness (MPa) at Loading Time: 10 (s) | | |
|:---:|:---:|:---:|:---:|
| | High Temperature | Intermediate Temperature | Low Temperature |
| V | 0.01 | 0.04 | 0.31 |
| CR | 0.01 | 0.06 | 0.43 |
| SA | 0.01 | 0.05 | 0.36 |
| CE | 0.00 | 0.03 | 0.31 |
| RE | 0.01 | 0.04 | 0.34 |
| SECR | 0.01 | 0.07 | 0.49 |
| CECR | 0.01 | 0.05 | 0.39 |
| RECR | 0.01 | 0.06 | 0.44 |

**Table 5.** *Cont.*

| Asphalt ID | Stiffness (MPa) at Loading Time 5 (s) | | |
|---|---|---|---|
| | **High Temperature** | **Intermediate Temperature** | **Low Temperature** |
| V | 0.01 | 0.07 | 0.47 |
| CR | 0.02 | 0.11 | 0.7 |
| SA | 0.01 | 0.08 | 0.54 |
| CE | 0.01 | 0.06 | 0.48 |
| RE | 0.01 | 0.07 | 0.5 |
| SECR | 0.02 | 0.13 | 0.83 |
| CECR | 0.01 | 0.09 | 0.61 |
| RECR | 0.02 | 0.1 | 0.73 |
| Asphalt ID | Stiffness (MPa) at Loading Time: 1 (s) | | |
| | **High Temperature** | **Intermediate Temperature** | **Low Temperature** |
| V | 0.05 | 0.32 | 1.48 |
| CR | 0.07 | 0.37 | 1.93 |
| SA | 0.05 | 0.31 | 1.71 |
| CE | 0.04 | 0.24 | 1.7 |
| RE | 0.05 | 0.28 | 1.71 |
| SECR | 0.08 | 0.42 | 2.51 |
| CECR | 0.06 | 0.34 | 1.87 |
| RECR | 0.07 | 0.38 | 2.35 |
| Asphalt ID | Stiffness (MPa) at Loading Time: 0.5 (s) | | |
| | **High Temperature** | **Intermediate Temperature** | **Low Temperature** |
| V | 0.08 | 0.42 | 2.58 |
| CR | 0.13 | 0.56 | 3.48 |
| SA | 0.09 | 0.47 | 3.09 |
| CE | 0.07 | 0.4 | 3.23 |
| RE | 0.08 | 0.43 | 3.16 |
| SECR | 0.15 | 0.67 | 4.17 |
| CECR | 0.1 | 0.49 | 3.45 |
| RECR | 0.11 | 0.58 | 4.11 |
| Asphalt ID | Stiffness (MPa) at Loading Time: 0.1 (s) | | |
| | **High Temperature** | **Intermediate Temperature** | **Low Temperature** |
| V | 0.3 | 1.23 | 7.81 |
| CR | 0.42 | 1.64 | 9.24 |
| SA | 0.35 | 1.44 | 8.67 |
| CE | 0.26 | 1.31 | 9.93 |
| RE | 0.3 | 1.39 | 9.43 |
| SECR | 0.46 | 1.88 | 11.04 |
| CECR | 0.37 | 1.57 | 9.17 |
| RECR | 0.41 | 1.8 | 11.31 |

## 4. Conclusions

This research investigated the effects of three warm mix additives on the physical properties of rubberized asphalt binder by performing physical tests, namely the penetration, softening point, penetration index, penetration viscosity number, storage stability, ductility, viscosity test, and stiffness modulus. Based on the research results, the researchers draw the following conclusions.

- Enhanced asphalt binders, resulting from modifications, display increased levels of hardness. Notably, the binder that underwent modification through the introduction of crumb rubber and Sasobit showcases the most pronounced hardness.
- All rubberized asphalt binders modified with warm mix asphalt (WMA) techniques manifest elevated values of softening points in comparison to the original asphalt binder. Among these alternatives, the rubberized asphalt binder that incorporates Sasobit stands out with the highest softening point.
- The penetration index, penetration viscosity number, and loss on heating of crumb rubber with WMA additives closely resemble those of the virgin asphalt binder.
- The modified asphalt binders exhibit commendable stability over the course of storage.
- With the exception of the rubberized asphalt binder adjusted with Rediset, all other modified asphalt binders adhere to the criteria set by the ductility test.
- All pairings of rubberized asphalt binder and WMA additives contribute to the reduction of viscosity values. Notably, the rubberized asphalt binder with Sasobit showcases superior workability compared to its counterparts employing alternative WMA additives.
- Among the assortment of asphalt binders, the combination that involves crumb rubber and Sasobit-modified binder presents the highest stiffness modulus values under conditions of intermediate and high temperatures.
- When subjected to lower temperatures and briefer loading periods, the blend of Rediset and crumb rubber produces the most significant stiffness modulus values.

The findings suggest that employing crumb rubber with Sasobit could be effectively implemented in nations with hot weather conditions, whereas the combination of crumb rubber and Rediset could be suitable for countries experiencing colder climates. For forthcoming studies, it is recommended to center attention on examinations conducted at lower temperatures, specifically emphasizing tests like the Bending Beam Rheometer (BBR) and the Direct Tension Test (DTT). Furthermore, an augmentation of the investigation could involve the incorporation of a broader collection of warm-mix asphalt additives.

**Author Contributions:** Conceptualization, N.I.M.Y.; Methodology, M.B.; Validation, C.W.Y.; Investigation, M.B. and Z.H.A.-S.; Data curation, C.W.Y.; Writing—original draft, M.B.; Writing—review & editing, Z.H.A.-S. and S.R.O.A.; Visualization, M.A. and S.R.O.A.; Supervision, N.I.M.Y.; Funding acquisition, M.A. All authors have read and agreed to the published version of the manuscript.

**Funding:** This work was funded by Science and Engineering Research Center at Najran University with grant (NU/RCP/SERC/12/3).

**Institutional Review Board Statement:** Not applicable.

**Informed Consent Statement:** Not applicable.

**Data Availability Statement:** All data used in this research can be provided upon request.

**Acknowledgments:** The authors are thankful to the Deanship of Scientific Research under supervision of the Science and Engineering Research Center at Najran University for funding this work under the research centers funding program with grant (NU/RCP/SERC/12/3). The authors would like to thank Universiti Malaya for their continuous support.

**Conflicts of Interest:** The authors have no conflict of interest to declare.

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
