# Peer review of "Influence of Warm Mix Asphalt Additives on the Physical Characteristics of Crumb Rubber Asphalt Binders"

_applsci, doi:10.3390/app131810337_

Round 1

Reviewer 1 Report

1) Introduction part can be improvised by highlighting the research gap more precisely.

2) One paragraph can be added why specifically crumbed rubber is selected as the additives with asphalt binders.

3) How this study help in achieving sustainability can be discussed in the results and discussion part so that it will be very useful for the readers.

4) Try to Highlight the limitations of the study.

5) Highlight the process of the study by means of flow chart or process chart or in the form of graphical abstract .

1) Many statements are too lengthy and very difficult to understand.Can you make it in simple language.

Author Response

Review Report Comments for Reviewer 1

Manuscript ID:  applsci-2569390

Thank you for your positive, fruitful comments and suggestions, which have improved our manuscript's quality. Please find below is the revision report for your attention and perusal. The responses have been arranged based on your feedback in the review process.

Comments

Amendments

Location of the additional write up

1) Introduction part can be improvised by highlighting the research gap more precisely.

Thank you.

The gap in the research have been added and modified.

-           

2) One paragraph can be added why specifically crumbed rubber is selected as the additives with asphalt binders.

Thank you for your comment.

The answer for selecting the CR binder, it can be found in lines 63-70.

-           

3) How this study help in achieving sustainability can be discussed in the results and discussion part so that it will be very useful for the readers.

Thank you for your valuable comment.

To discuss sustainability in the result section, it needs to conduct some specific tests such as measuring the emission during the mixing and compaction process. Unfortenly, that test is not available now.

The authors will take your comment for future work.

-           

4) Try to Highlight the limitations of the study.

Thank you for your comment.

The limitations for the warm mix asphalt have been added.

-           

5) Highlight the process of the study by means of flow chart or process chart or in the form of graphical abstract .

Thank you for your valuable comment. A flowchart has been added to the manuscript.

Figure 3

Thank you for your kind attention and cooperation. We appreciate all your comments and feedback.

Reviewer 2 Report

This work is very informative and the efforts made by the authors are worthy of recognition. There are some details that should be revised before the paper is accepted.
1. in the references, there are fewer citations of the authors' most recent studies from 2022-2023, which should be added.
2. check the unit of vertical coordinate of Fig. 3 is correct.
3. there is no citation about Figures 1 and 7 in the main text.
4. there is no citation in the main text regarding Table 1 and.

Author Response

Review Report Comments for Reviewer 2

Manuscript ID:  applsci-2569390

Thank you for your positive, fruitful comments and suggestions, which have improved our manuscript's quality. Please find below is the revision report for your attention and perusal. The responses have been arranged based on your feedback in the review process.

Comments

Amendments

Location of the additional write up

1. in the references, there are fewer citations of the authors' most recent studies from 2022-2023, which should be added.

Your comment is appreciated.

As the reviewer suggestion, a recent studies have been added in introduction and results sections.

-           

2. check the unit of vertical coordinate of Fig. 3 is correct.

Thank you for your comment

The unit is correct.

-           

3. there is no citation about Figures 1 and 7 in the main text.

Thanks for the kind observation.

For figure 1, it has been added.

For figure 7, it can be found in line 340-341.

Lines 168-169

4. there is no citation in the main text regarding Table 1 and.

Thanks again for the comment.

For table 1, it has been added.

Lines 169-170.

Thank you for your kind attention and cooperation. We appreciate all your comments and feedback.

Reviewer 3 Report

The manuscript is titled "EFFECTS OF WARM MIX ASPHALT ADDITIVES ON THE PHYSICAL PROPERTIES OF CRUMB RUBBER ASPHALT BINDERS." is required a few modifications.

1. Revise the Introduction

2. Write specific objectives in the last part of the introduction.

3. Specific literature needs to incorporate.

4. Table 5 is not mentioned ( line no. 359).

5. Results and discussion also need to discuss with other author's work.

6. Rewrite conclusion

7. Future scope is missing

A few grammatical corrections need to address. 

Author Response

Review Report Comments for Reviewer 3

Manuscript ID:  applsci-2569390

Thank you for your positive, fruitful comments and suggestions, which have improved our manuscript's quality. Please find below is the revision report for your attention and perusal. The responses have been arranged based on your feedback in the review process.

Comments

Amendments

Location of the additional write up

1. Revise the Introduction

Thanks for the valuable comment.

Parts of introduction have been revised and more specific information has been added.

Introduction

2. Write specific objectives in the last part of the introduction.

Your comment is appreciated. The objective of the study has been added.

Lines 150-153

3. Specific literature needs to incorporate.

Your invaluable comment is appreciated. More studies have been added in the introduction.

Lines 88-97

Lines 132-145

Lines 148-153

4. Table 5 is not mentioned (line no. 359).

That was a useful comment from the reviewer. For table 5, it has been added.

Lines 398-399.

5. Results and discussion also need to discuss with other author's work.

Thanks for the valuable comment. The authors have been added more discussion in sections 3.2, 3.4, and 3.7.

Lines 282-285

Lines 320-324

Lines 369-373

6. Rewrite conclusion

Thanks for your comment.

The conclusion has been modified.

Conclusion

7. Future scope is missing

Thanks for the valuable comment. The authors have been added forthcoming studies.

Lines 437-441

Thank you for your kind attention and cooperation. We appreciate all your comments and feedback.

Reviewer 4 Report

In the article titled “Influence of Warm Mix Asphalt Additives on the physical characteristics of Crumb Rubber asphalt binders”, the author conducted research on Warm mix asphalt used as an additive and its influence on the characteristics of Crumb Rubber asphalt binders. After a careful review of the article, I have found that article covers the effect of adding three different additives Sasobit, Cecabase RT, and  Rediset WMX on the physical properties of rubberized asphalt binders. This objective has been done by incorporating these additives into eight different asphalt binders. I consider that the manuscript is relevant to the journal, but its quality and content must be improved. Specifically, I have the following comments:

1.     The grammatical and spelling aspects of the paper need to be improved.

2.     Abstract: The abstract is not technically written and the novelty of the study along with a conclusion of the research. The quantitative information is missing in the abstract. The abstract must include quantitative information.

3.     Keywords: add new keywords, NOT FROM TITLE

4.     Introduction: Last para of introduction must contain the objectives and novelty of your work, but it is entirely missing in the article. The research gap or need for the study is not clearly mentioned. And a review of the results of a similar study needs to be included in this section. I suggest more literature should be added regarding the usage of these additives individually in the crumb rubber asphalt binders. The objective of this study is NOT clearly mentioned in the introduction part. It must include clear objectives of this study.

5.     Line 65. Please use full form for abbreviation in its first appearance only.

6.     Authors should mention some limitations and highlights of Warm Mix Asphalt additives activity in the introduction with standard references.

7.     There should be uniform spaces (single/none) between numbers and units (°C, mg/L etc.). Kindly check all.

8.     Section 2 The author covers the detailing of the different materials used in the production of samples along with the physical properties of the material and different combination ratios of additives used in the samples. The detailing of the procedure for the production of testing samples is not significantly mentioned in this paper.

9.     In Results and Discussion section, the discussion on figures should be improved.

10.  Section 3 In this section the error bar needs to be present in order to support the results and conclusion of the research. Error bars should be present in all figures present in the result sections. All more literature needs to present in the discussion section to support the results.

11.  The use of subscripts and superscripts needs to be reviewed throughout the manuscript.

12.  Why authors have used Warm Mix Asphalt additives as ingredient material? Kindly justify it?

13.  In section 3.1 Penetration helps in determining the hardness of asphalt. How the hardness is improved by the combination of two or more additives needs to be explained in this section.

14.  In determining the influence of the incorporation of different additives on the softening point of crumb rubber, there is very limited literature supporting the result of softening point of rubber crump. More literature is required to support the current study.

15.  There is no comparison of the result with the previous study in the section on Temperature susceptibility, so the previous study needs to be discussed in this section.

16.  Similarly, a discussion of the previous study is not mentioned in section 3.7. So previous studies need to incorporate to support the result data.

17.  All the conclusions should be numbered or in bullet point form. Also, authors need to rephrase the conclusions.

Author Response

Review Report Comments for Reviewer 4

Manuscript ID: applsci-2569390

Thank you for your positive, fruitful comments and suggestions, which have improved our manuscript's quality. Please find below is the revision report for your attention and perusal. The responses have been arranged based on your feedback in the review process.

Comments

Amendments

Location of the additional write up

1.     The grammatical and spelling aspects of the paper need to be improved.

Thanks for the valuable comment.

we have fixed and proofreading the manuscript to eliminate all such errors.

-

2.     Abstract: The abstract is not technically written and the novelty of the study along with a conclusion of the research. The quantitative information is missing in the abstract. The abstract must include quantitative information.

Thank you for the suggestions.

The abstract has been revised.

Abstract  

3.     Keywords: add new keywords, NOT FROM TITLE

Thanks for the kind observation and suggestions for improvement. The authors have added more keywords.  

Keywords

4.     Introduction: Last para of introduction must contain the objectives and novelty of your work, but it is entirely missing in the article. The research gap or need for the study is not clearly mentioned. And a review of the results of a similar study needs to be included in this section. I suggest more literature should be added regarding the usage of these additives individually in the crumb rubber asphalt binders. The objective of this study is NOT clearly mentioned in the introduction part. It must include clear objectives of this study.

Thank you for the suggestions.

The objectives of the study have added in the introduction section.

Lines 150-153

5.     Line 65. Please use full form for abbreviation in its first appearance only.

Thank you for your comments.

All abbreviations have been mentioned for the first time in the manuscript.

-           

6.     Authors should mention some limitations and highlights of Warm Mix Asphalt additives activity in the introduction with standard references.

Thank you for your observation. The authors have added the limitation of the Warm Mix Asphalt.

Lines 88-97

7.     There should be uniform spaces (single/none) between numbers and units (°C, mg/L etc.). Kindly check all.

Thank you for your valuable comment.

All have been corrected in the manuscript.

-           

8.     Section 2 The author covers the detailing of the different materials used in the production of samples along with the physical properties of the material and different combination ratios of additives used in the samples. The detailing of the procedure for the production of testing samples is not significantly mentioned in this paper.

Thanks for the observation.

Some of the test processes are not mentioned because the authors mention the standard for the tests since the test process is easy to apply by the standards guide.

-           

9.     In Results and Discussion section, the discussion on figures should be improved.

Thanks for the suggestion. The authors have added some discussion for the results sections.

Section 3.2, 3.4 and 3.7

10.  Section 3 In this section the error bar needs to be present in order to support the results and conclusion of the research. Error bars should be present in all figures present in the result sections. All more literature needs to present in the discussion section to support the results.

Thank you for your comment.

The error bars have been added for all result figures, as the reviewer suggestion.

Figures 4-9

11.  The use of subscripts and superscripts needs to be reviewed throughout the manuscript.

Thanks for your comment.

subscripts and superscripts have been corrected in the manuscript.

-           

12.  Why authors have used Warm Mix Asphalt additives as ingredient material? Kindly justify it?

Thanks for your valuable comment. the authors chose the WMA additives based on the previous studies. For general purposes, the WMA additives can reduce the mixing and compaction temperature which leads to a decrease the energy consumption and greenhouse gases. For technical purposes, WMA additives can improve the workability of the CR asphalt binder.

-           

13.  In section 3.1 Penetration helps in determining the hardness of asphalt. How the hardness is improved by the combination of two or more additives needs to be explained in this section.

Thank you for your comment.

The purpose of using warm mix asphalt for the rubberized asphalt is to improve the workability without major effect on the hardness of the rubberized asphalt.

-           

14.  In determining the influence of the incorporation of different additives on the softening point of crumb rubber, there is very limited literature supporting the result of softening point of rubber crump. More literature is required to support the current study.

Thank you for your valuable comment. The authors have added more discussion in the section 3.2.

Lines 820-285

15.  There is no comparison of the result with the previous study in the section on Temperature susceptibility, so the previous study needs to be discussed in this section.

Thanks for your suggestion. The authors have added more discussion in the section 3.4.

Lines 320-324

16.  Similarly, a discussion of the previous study is not mentioned in section 3.7. So previous studies need to incorporate to support the result data.

Thanks for your comment. The authors have added more discussion in the section 3.7.

Lines 369-373

17.  All the conclusions should be numbered or in bullet point form. Also, authors need to rephrase the conclusions.

Thanks for your comment.

The conclusion has been modified and put it in bullet point form.

conclusion

Thank you for your kind attention and cooperation. We appreciate all your comments and feedback.

Round 2

Reviewer 4 Report

The manuscript has been well-revised. All the comments have been considered in the revised version. Therefore, I recommend accepting it in its current form.